# The Effect of Structured Exercise Compared with Education on Neuropathic Signs and Symptoms in People at Risk of Neuropathic Diabetic Foot Ulcers: A Randomized Clinical Trial

**DOI:** 10.3390/medicina58010059

**Published:** 2021-12-30

**Authors:** Byron M. Perrin, Jessica Southon, Jane McCaig, Isabelle Skinner, Timothy C. Skinner, Michael I. C. Kingsley

**Affiliations:** 1La Trobe Rural Health School, La Trobe University, Bendigo 3550, Australia; b.perrin@latrobe.edu.au (B.M.P.); J.McCaig@latrobe.edu.au (J.M.); T.Skinner@latrobe.edu.au (T.C.S.); 2Castlemaine Health, Castlemaine 3450, Australia; JSouthon@castlemainehealth.org.au; 3Active Podiatry, Bendigo 3550, Australia; 4Fusion Physiotherapy, 86 McIvor Hwy, Bendigo 3550, Australia; 5Centre for Rural and Remote Health, James Cook University, Townsville 4810, Australia; Isabelle.skinner@jcu.edu.au; 6Holsworth Research Initiative, La Trobe University, Bendigo 3550, Australia; 7Department of Exercise Sciences, University of Auckland, Auckland 1023, New Zealand

**Keywords:** exercise, diabetic polyneuropathy, cutaneous innervation, pilot study, randomized controlled trial, diabetes

## Abstract

*Background and Objecti**ves*: Lifestyle interventions such as exercise prescription and education may play a role in the management of peripheral neuropathy in people with diabetes. The aim of this study was to determine the effect of undertaking an exercise program in comparison with an education program on the signs and symptoms of peripheral neuropathy in people with diabetes at risk of neuropathic foot ulceration. *Materials and Methods:* Twenty-four adult participants with diabetes and peripheral neuropathy were enrolled in this parallel-group, assessor blinded, randomised clinical trial. Participants were randomly allocated to one of two 8-week lifestyle interventions, exercise or education. The primary outcome measures were the two-part Michigan Neuropathy Screening Instrument (MNSI) and vibratory perception threshold (VPT). Secondary outcome measures included aerobic fitness, balance and lower limb muscular endurance. *Results:* Participants in both lifestyle interventions significantly improved over time for MNSI clinical signs (MD: −1.04, 95% CI: −1.68 to −0.40), MNSI symptoms (MD: −1.11, 95% CI: −1.89 to −0.33) and VPT (MD: −4.22, 95% CI: −8.04 to −0.40). Although the interaction effects did not reach significance, changes in values from pre to post intervention favoured exercise in comparison to control for MNSI clinical signs (MD −0.42, 95% CI −1.72 to 0.90), MNSI clinical symptoms (MD −0.38, 95% CI −1.96 to 1.2) and VPT (MD −4.22, 95% CI −12.09 to 3.65). *Conclusions:* Eight weeks of exercise training or lifestyle education can improve neuropathic signs and symptoms in people with diabetes and peripheral neuropathy. These findings support a role for lifestyle interventions in the management of peripheral neuropathy.

## 1. Introduction

Peripheral neuropathy is the progressive degeneration and dysfunction of sensory, autonomic and motor nerve fibres and nerves with the most common form of peripheral neuropathy being diabetic sensorimotor polyneuropathy [1]. Peripheral neuropathy is one of the most serious and common complication of diabetes mellitus, affecting up to 50% of those with diabetes mellitus [2], and is a well-established risk factor for diabetes-related foot ulceration and lower limb amputations. Lower limb ulceration significantly impacts upon quality of life, and places a significant financial burden on households and health systems [3,4]. Almost 50% of those who have a lower limb ulceration will be at risk of future ulceration and amputation [1] and five year survival rates for amputation are poor, with mortality rates reported to be 43–74% [5]. Recent data suggests that diabetes-related foot disease causes ~2% of the global disability burden and ~1–2% of all healthcare costs [6,7], and that the proportion of that burden for peripheral neuropathy is substantial [6].

Currently, there are limited treatment options for peripheral neuropathy, and treatment is limited to maintaining very strict glucose control and pain relief for painful neuropathy [8]. While the positive effects of exercise on general glucose control and physical function in this population has been established [9], there is an emerging body of evidence to support lifestyle interventions such as exercise prescription as an effective treatment for people with existing peripheral neuropathy [10]. In addition to being safe and feasible, exercise may directly influence the signs and symptoms of peripheral neuropathy in this population [10,11,12,13]. However, trials are restricted to mainly tertiary level settings and vary with respect to the intervention prescribed, control group activity, outcome measures, and overall quality, and there is limited high quality evidence from community-based trials demonstrating a clinically beneficial effect [12,13,14].

A randomised clinical trial is required to test the hypothesis that exercise has a beneficial effect for peripheral neuropathy when compared to commonly prescribed diabetes lifestyle education in a pragmatic, community-based setting. To this end, the aim of this study was to determine the effect of undertaking an exercise program in comparison with an education program on the signs and symptoms of peripheral neuropathy in people with diabetes at risk of neuropathic foot ulceration.

## 2. Material and Methods

This parallel-group, assessor blinded, randomised clinical trial was registered with the Australia and New Zealand Clinical Trials Registry (ACTRN12614000048684). Ethical approval was gained from the La Trobe University Human Ethics Committee (HEC13-042). The trial is reported according to the Consolidated Standards of Reporting Trials (CONSORT) [15].

Participants with diabetes and peripheral neuropathy were recruited in two ways. Firstly, a targeted approach was used that re-identified a small number of de-identified enrolled participants from an existing epidemiological study of the foot-health of people with diabetes in regional and rural Australia. Invitations to participate were sent to 126 participants. Secondly, potential participants were informed of the project through advertisements in the local newsletters and private podiatry clinical practices.

Participants were randomised to receive an exercise intervention or an education program. Allocation to either the exercise or education groups was achieved using a computer generated random number sequence [15]. The allocation sequence was generated and held by an external person not directly involved in the trial. The allocation sequence was concealed from the researchers assessing the primary outcome measure of the study. It was not possible to blind the participants to group allocation due to the nature of the activity of the groups (group exercise or group education).

Inclusion criteria were aged 18 years or older, clinical diagnosis of diabetes confirmed by General Practitioner, peripheral neuropathy as defined by a Michigan Neuropathy Screening Instrument (MNSI) score > 2 (examination component) [16], and approval from the participants General Practitioner to participate. Exclusion criteria were: diagnosis of painful neuropathy, use of medication for the treatment of peripheral neuropathy, active foot ulcer, inability to understand instructions and established exercise based guidelines, inability to understand instructions, serious cardiac pathology, musculoskeletal problems that would seriously limit exercise, open wounds on the weight-bearing surface of the feet, inability to ambulate independently, stroke or other central nervous system pathology, stage 2 hypertension (resting blood pressure: systolic > 160 mmHg or diastolic > 100 mmHg), atrial fibrillation and/or pacemaker [17].

Two 1-h assessments were conducted prior to randomisation and allocation to groups (baseline). The first assessment focussed on screening for foot-health related exclusion criteria, confirming inclusion criteria, measuring the primary outcome measures, measuring basic demographic and diabetes related variables, assessing footwear and administering the International Physical Activity Short Questionnaire (to determine baseline levels of self-reported exercise) [18]. At the second appointment exercise-based exclusion criteria was assessed and secondary variables were measured including basic anthropometry [19], physical function tests of aerobic fitness [20], tandem stance as a means of assessing standing balance and postural steadiness [21], and lower limb muscular endurance [22]. Anthropometric variables and the primary and secondary outcomes were repeated by the same assessors after the exercise and education programs were completed (post intervention).

The 8-week exercise program was based on American Diabetes Association and American College of Sports Medicine recommendations, was consistent with previous research with this patient group and was comprised of three approximately 1 h exercise sessions per week that included both aerobic and strengthening components [11,17,20]. The program was individually prescribed with full consideration of medical history and was undertaken in a group environment of no more than eight participants. All exercise sessions were preceded by warm-up and stretching. The intended prescription for the aerobic component was to progress from 10 min at 50% heart rate reserve to 50 min at 70% heart rate reserve indicative of moderate-intensity exercise (calculated by proportion of the difference between resting heart rate and age-predicted maximal heart rate). The strengthening component was prescribed to start at 2 sets of 10 repetitions gradually progressing to 3 sets of 15 repetitions to maintain rating of perceived exertion in a moderate range (7–8 out of 10) for eight exercises (sit-to-stand/squat, chest press, step ups, lateral pulldown, heel raises, upright rows, bicep curls, shoulder press). The exercise intervention was individually prescribed and supervised by an Accredited Exercise Physiologist.

The comparative group was offered a group education program underpinned by the “Diabetes Conversations” program, which was developed by the International Diabetes Federation based on international guidelines [23]. It was an interactive, group-based program focused on key areas of living with diabetes that include diet, medication, physical activity and other health-related behaviours. Education was provided to the comparison group because best practice management for people with type 2 diabetes includes education on lifestyle choices (mainly with the aim to assist with blood glucose control). The exercise and education programs were congruent with guidelines for the Medicare funded Type 2 Diabetes Allied Health Group Program, an Australian federal health cost subsidy scheme [24]. The exercise and education groups occurred during the same 8-week period.

The primary outcomes measured were the two components of the Michigan Neuropathy Screening Instrument (MNSI) [16]. The first component of the MNSI is a self-report, 13-item questionnaire (two items were excluded relating to non-neuropathy factors) that investigates signs and symptoms of peripheral neuropathy such as foot sensation, foot pain and foot temperature sensation [16]. The second component is a short clinical examination of the foot focusing on foot appearance, ulceration, ankle reflexes and vibration perception at the hallux, with the final result being a score out of 8 [16]. The MNSI clinical signs component is validated to assess for signs of peripheral neuropathy, where higher scores represent worse loss of sensation [16]. Vibratory perception threshold (VPT) was also measured (maximum score 50 volts). VPT is a validated tool to diagnose loss of protective sensation related to peripheral neuropathy and can be used as a predictive tool for foot morbidity [25]. Secondary outcome measures consisted of physical function tests of aerobic fitness [20], balance (tandem stance) [21] and lower limb muscular endurance (sit-to-stand) [22]. Aerobic fitness was measured using a progressive submaximal cycle test protocol to estimate maximum oxygen uptake using metabolic calculations for leg cycling [20].

Statistical analysis was completed using IBM SPSS software IBM SPSS software (Version 22.0. IBM Corp., Armonk, NY, USA). Group data were expressed as mean ± standard deviation for ratio data or proportions for nominal data. Statistical significance was set at *p* < 0.05. Participant characteristics were compared at baseline using independent sample t-tests for ratio data or chi-square for nominal data (Table 1). Mixed-model repeated measured ANOVAs (between factor: intervention group; within factor: timing) were undertaken for anthropometry, primary and secondary outcome variables. Mauchly’s test was reviewed and Greenhouse–Geisser correction was applied if the assumption of sphericity was violated. If a significant *p*-value was identified for the interaction effect (intervention X timing), intervention was deemed to influence the pattern of response. The main effects of intervention and timing were consulted if the interaction effect did not reach significance.

## 3. Results

The progression of participants through the trial is presented in Figure 1. In total, 38 individuals volunteered to take part in the study. Of the 126 potential participants invited from the existing database, 19 (15%) responded and were assessed as eligible based on the inclusion criteria following the primary baseline assessment. A further 19 (73% of respondents from the public advertisement) were included after the public recruitment. After the inclusion and exclusion criteria were implemented, 24 participants were randomly allocated to either exercise (*n* = 12) or education (*n* = 12). Two participants from the exercise group did not complete the post intervention testing and were unable to be contacted to assess any outcomes post intervention.

Participants in both groups had similar anthropometric and diabetes-related characteristics at baseline (Table 1). MNSI clinical signs score were higher in exercise compared to education (*p* = 0.04) indicative of slightly worse baseline peripheral neuropathy in exercise at baseline. Overall, the sample was older (71.1 ± 10.3 years), 54.2% male, 95.8% type 2 diabetes, reported low levels of self-reported exercise (47.8% not adhering to exercise guidelines) and were aerobically unfit (maximum oxygen uptake: 19.8 ± 3.9 mL·kg^−1^·min^−1^).

Table 2 displays the mean difference in primary and secondary outcome measures for the combined cohort and the exercise and education groups. The exercise and education interventions resulted in similar patterns of change (Figure 2) in neuropathic signs (interventionXtiming interaction effect: F_(1,20)_ = 0.46, *p* = 0.51, partial Eta^2^ = 0.02, MD −0.42 95% CI −1.72 to 0.90), neuropathic symptoms (interventionXtiming effect: F_(1,20)_ = 0.26, *p* = 0.61, partial Eta^2^ = 0.01; MD −0.38 95% CI −1.96 to 1.20) and VPT (interventionXtiming effect: F_(1,20)_ = 1.33, *p* = 0.26, partial Eta^2^ = 0.06; MD −4.22 95% CI −12.09 to 3.65).

Nevertheless, participants in both lifestyle interventions significantly improved neuropathic signs (timing effect: F_(1,20)_ = 11.47, *p* = 0.003, partial Eta^2^ = 0.36, MD −1.04 95% CI −1.68 to 0.90), neuropathic symptoms (timing effect: F_(1,20)_ = 8.79, *p* = 0.008, partial Eta^2^ = 0.31, MD −1.11 95% CI −1.89 to −0.33) and VPT (timing effect: F_(1,20)_ = 5.29, *p* = 0.03, partial Eta^2^ = 0.21, MD −4.22 95% CI −8.04 to −0.40).

Consistent with the primary outcome measures, the pattern of change in maximum oxygen uptake was similar when comparing exercise and education (interventionXtiming interaction effect: F_(1,20)_ = 0.75, *p* = 0.40, partial Eta^2^ = 0.04). When combined, participants in both lifestyle interventions significantly improved maximum oxygen uptake over time (timing effect: F_(1,20)_ = 5.29, *p* = 0.03, partial Eta^2^ = 0.24). Similarly, sit to stand results demonstrated no difference in the pattern of change between groups (interventionXtiming interaction effect: F_(1,20)_ = 0.03, *p* = 0.87, partial Eta^2^ = 0.001); however, participants in both lifestyle interventions improved over time (timing effect: F_(1,20)_ = 8.58, *p* = 0.01, partial Eta^2^ = 0.31). The interaction and time effects for the tandem stance did not reach significance (interventionXtiming effect: F_(1,20)_ = 0.03, *p* = 0.86, partial Eta^2^ = 0.002; timing effect: F_(1,20)_ = 1.5, *p* = 0.24, partial Eta^2^ = 0.07). See Table 2 for the mean differences for the secondary variables.

Mean attendance at the exercise and educations sessions were acceptable at >80%. No adverse events such as foot ulceration, medical incidents or other events such as increased pain were reported during the trial. The duration and intensity of aerobic component ranged from 7 to 20 min in week 1 to 15 to 45 min in week 8 at the prescribed relative exercise intensity (50–70% heart rate reserve). The strengthening component of the exercise program began with 2 sets of 10 repetitions on each exercise and progressed to 3 sets of 10–15 repetitions at a RPE of 7–8 out of 10.

## 4. Discussion

The main finding of this study was that a lifestyle intervention over an 8-week period (exercise or education) improved neuropathic signs and neuropathic symptoms, demonstrating that exercise or education lifestyle interventions can in the short-term improve signs and symptoms of peripheral neuropathy.

For the three primary outcome measures there was a trend for greater improvements in neuropathic signs and symptoms that favoured exercise. The largest effect in the mean difference between exercise and education for the pre to post intervention change was for VPT, which is the most robust of the primary outcome measures to detect loss of protective sensation [25]. This trend was also seen recently in a South Asian Indian population [26], and the effectiveness of exercise training to reduce neuropathic signs and symptoms has been demonstrated by other researchers using other neuropathic measures [27,28,29]. Although the mechanisms by which exercise training can improve neurological function remain to be fully elucidated, a number of well-established physiological responses to exercise might at least partially explain these findings. Exercise training can enhance mitochondrial function, improve endogenous antioxidant defence and attenuate oxidative stress, which is a process that has consistently been associated with neural dysfunction [11,30,31]. Additionally, exercise training increases nitric oxide production and vasodilation of neural structures [32], thereby reducing hypoxic conditions in nerves, which can cause necrosis. Furthermore, physical exercise stimulates innervation of neuronal activity [33]. Branching of cutaneous nerves has also been observed after exercise in this population [29].

Although the pattern of change in maximum oxygen uptake, balance or lower leg strength did not differ between intervention groups, both exercise and education improved maximum oxygen uptake and balance. In accordance with previous studies, it was expected that those in the exercise group would improve physical function more than those in the education group [9,29]. However, the participants in this study were generally older with very low baseline levels of self-reported physical activity and physical function. It is possible, therefore, that the dose of exercise training undertaken by the exercise group was not sufficient enough over an 8-week duration to improve maximum oxygen uptake and balance to an extent that was greater than the education group. Although speculative, it is possible that maximum oxygen uptake and balance improved over time for the education and exercise groups because participants in both groups increased their physical activity as a result of the lifestyle intervention they received.

The results of this study demonstrate that health professionals should offer exercise or education lifestyle interventions for people with diabetes and peripheral neuropathy. Both programs can provide health benefits to people with diabetes. Lifestyle interventions may reduce reliance on medications through metabolic benefits associated with body mass control (e.g., blood pressure, lipoprotein balance) as well as resulting in musculoskeletal functional benefits [34]. Furthermore, the results of this study are consistent with previous findings showing that with adequate baseline screening and participant selection adverse events are unlikely and the risk of further ulceration in response to weight bearing exercise is low [11]. This study has shown that it is practical, possible and safe to deliver an exercise and education programs that might be useful in a regional community setting. Utilising government funding initiatives like the national Australian Type 2 Diabetes Allied Health Group Program may lessen the financial burden of similar programs to patients and engage community providers of exercise and education interventions [35].

Recruitment for this study used an existing database for an epidemiological study based in regional and rural Australia [36]. Previous research suggests that this sample comes from a background of lower socio-economic position, higher levels of general morbidity and reduced access to health care services compared with those living in metropolitan areas [36,37,38]. A useful finding of this study was that the initial mail-out to potentially eligible participants yielded a 19.6% response rate and a 76% eligibility rate for those respondents. In addition, participants were invited to attend via public advertisements in the local newspaper, with acceptable eligibility rates of 73%. This recruitment approach was effective and can be used as a guide for practitioners who might wish to target exercise interventions for this group of people and for future research design.

A limitation in this study was the relatively small sample size and the inability to detect a statistically significant difference between the intervention groups. However, a trend emerged, and effect sizes have been generated to inform an a priori sample size calculation for a definitive randomized controlled trial on the effect of exercise on peripheral neuropathy in similar populations. Due to the short intervention duration, Hb1Ac measurements were not included in this study, and as the diagnosis of peripheral neuropathy was based on clinical testing, it is possible that diabetes was not the cause of neuropathy in all participants. Although it was not expected that any short-term blood glucose benefit of exercise or education would affect neuropathic signs and symptoms, it is possible that variations blood glucose control could have had an influence. Without a group that received no intervention, there is no control group in this study. This might explain improvements over time in the primary and secondary outcome measures. For example, it is possible that through agreeing to participate in the study the participants were in a position to respond to a lifestyle intervention. The sample recruited was at the higher end of morbidity, with clinical loss of protective sensation and low levels of fitness, which might restrict the generalisability of the results to other populations. The benefit of exercise and lifestyle interventions in people with less severe peripheral neuropathy should be explored in future research. Utilisation of more sensitive diagnostic tools, such as nerve conduction studies, might allow the researchers to explore the mechanisms that explain the responses to lifestyle interventions [39].

## 5. Conclusions

This assessor-blinded, randomised clinical trial has shown that in this high-risk population with diabetes and peripheral neuropathy, eight weeks of exercise training or lifestyle education can improve neuropathic signs and symptoms in patients with diabetes. These findings demonstrate the potential positive effect of lifestyle interventions for patients with diabetes and peripheral neuropathy and can inform a priori power analysis for future, larger randomised controlled trials.

## Figures and Tables

**Figure 1 medicina-58-00059-f001:**
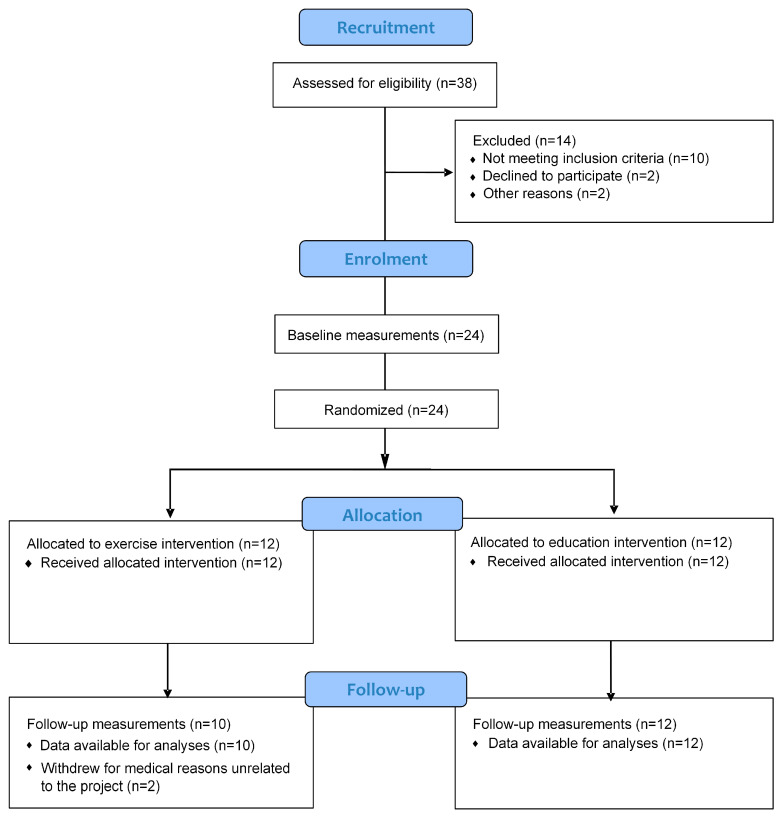
Flow diagram of the progress through the phases of the trial for two groups.

**Figure 2 medicina-58-00059-f002:**
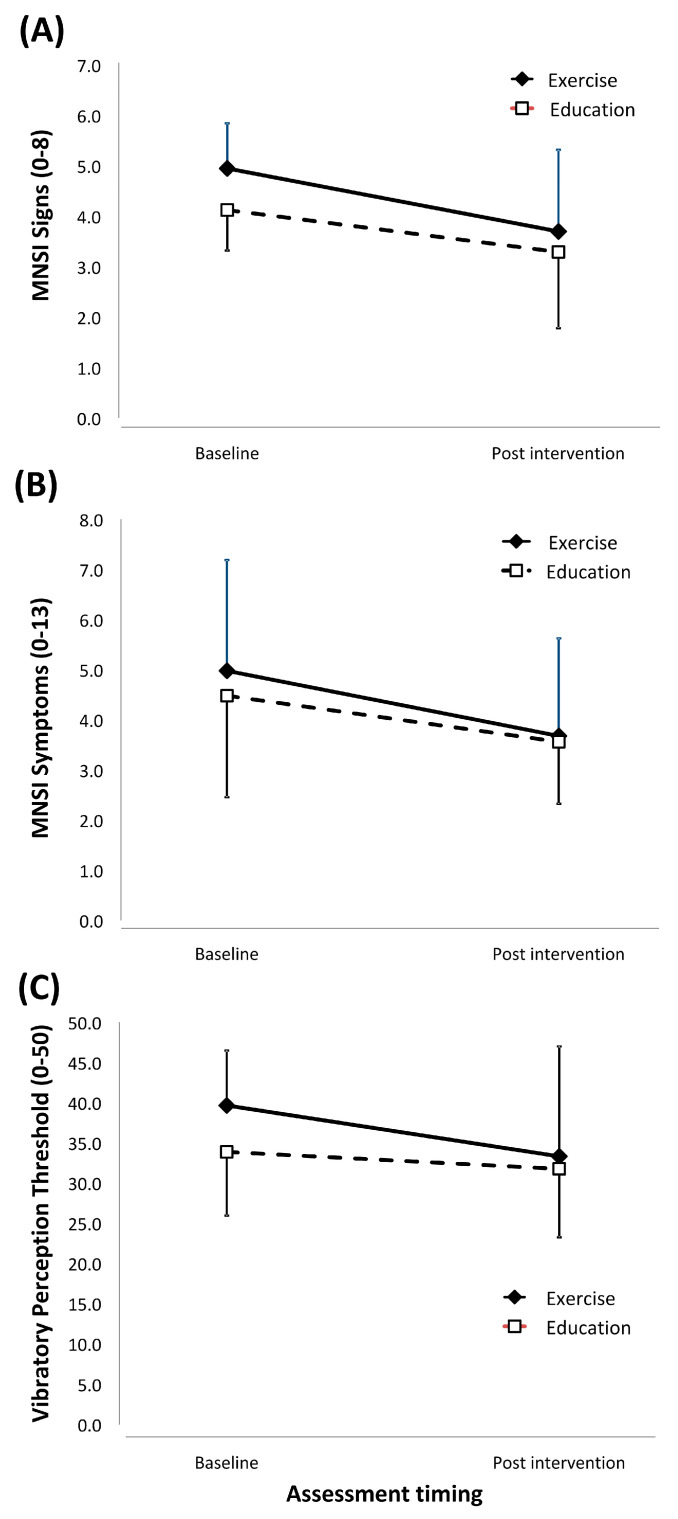
Graphical representation of change in primary outcome mean and standard deviation from baseline measurement to post-intervention measurement. ((**A**): MNSI Signs, (**B**): MNSI Symptoms, (**C**): Vibratory Perception Threshold).

**Table 1 medicina-58-00059-t001:** Participant characteristics at baseline.

Variable	Exercise (*n* = 12)	Education (*n* = 12)	All (*n* = 24)	*p* Value
Male sex (%)	58.3	50.0	54.2	0.68
Age (years)	70.1 (12.9)	72.2 (7.1)	71.1 (10.3)	0.63
Body Mass Index	35.6 (14.6)	31.9 (5.4)	33.6 (10.5)	0.46
Resting heart rate (bpm)	68.8 (11.4)	73.8 (9.2)	71.3 (10.5)	0.25
Resting systolic blood pressure (mm Hg)	137.8 (14.2)	132.1 (14.5)	134.9 (14.4)	0.35
Resting diastolic blood pressure (mm Hg)	74.2 (8.4)	77.4 (10.5)	75.8 (9.3)	0.41
Waist circumference (cm)	107.6 (19.1)	104.1 (10.6)	105.8 (15.1)	0.59
Type 2 diabetes (%)	91.7	100.0	95.8	0.31
Duration of diabetes (years)	9.8 (9.5)	14.2 (9.9)	12.0 (9.7)	0.28
Method of Control (%) Oral medication Insulin Diet Combination	16.7 16.7 41.7 25.0	41.7 16.7 25.0 16.7	29.2 16.7 33.3 20.8	0.58
Adhered to exercise recommendations *(%)	41.7	54.5	47.8	0.68
Michigan Neuropathy Screening Instrument. signs (/8) Michigan Neuropathy Screening Instrument. symptoms (/13) VPT (/50)	4.9 (0.9) 5.7 (2.5) 39.4 (6.2)	4.1 (0.8) 4.5 (2.0) 33.9 (7.9)	4.5 (0.9) 5.1 (2.3) 36.6 (7.5)	0.04 ^ 0.23 0.07
Balance: tandem stance (s)	37.0 (23.6)	40.6 (24.6)	38.8 (23.7)	0.72
Maximum oxygen uptake (mL·kg^−1^·min^−1^)	19.2 (3.2)	20.3 (4.6)	19.8 (3.9)	0.53
Sit to Stand (number 30 s)	16.4 (11.1)	14.8 (12.3)	15.6 (11.5)	0.75

Data presented as mean (standard deviation) for ratio data and proportions for nominal data. Differences between groups analysed using independent samples *t*-test for ratio variables and chi-square for nominal variables. ^ *p* < 0.05. * Based on MET. min/week calculated from the International Physical Activity Short Questionnaire.

**Table 2 medicina-58-00059-t002:** Mean (SD) and 95% confidence interval baseline and post intervention outcome scores for the combined cohort and the exercise and intervention groups.

Variable	Group	Baseline	Post Intervention	Mean Difference	95% CI
MNSI Signs (0–8)	Exercise Education	5.0 (0.9) 4.1 (0.8)	3.7 (1.6) 3.3 (1.5)	−0.42	−1.72 to 0.90
	Combined	4.5 (0.9)	3.5 (1.5)	−1.04 **	−1.68 to −0.40
MNSI Symptoms (0–13)	Exercise Education	5.0 (2.2) 4.5 (2.0)	3.7 (1.9) 3.6 (1.2)	−0.38	−1.96 to 1.2
	Combined	4.8 (2.1)	3.6 (1.6)	−1.11 **	−1.89 to −0.33
VPT (0–50)	Exercise Education	39.7 (6.8) 33.9 (7.9)	33.3 (13.7) 31.8 (8.5)	−4.22	−12.09 to 3.65
	Combined	36.8 (7.8)	32.5 (10.9)	−4.22 *	−8.04 to −0.40
Maximum oxygen uptake (mL·kg^−1^·min^−1^)	Exercise Education	19.2 (3.4) 19.5 (3.7)	22.2 (6.1) 21.5 (5.3)	1.64	−2.72 to 6.01
	Combined	19.3 (3.5)	21.5 (5.3)	2.18 *	0.18 to 4.18
Tandem Stance (s)	Exercise Education	37.2 (23.5) 38.8 (25.0)	39.8 (22.4) 42.3 (22.0)	−0.94	−11.44 to 9.57
	Combined	38.5 (23.7)	40.6 (21.7)	3.07	−12.21 to 8.35
30 s sit-to-stand (repetitions)	Exercise Education	15.4 (12.8) 15.1 (10.6)	22.2 (12.0) 21.2 (11.3)	0.71	−8.60 to 10.01
	Combined	15.2 (11.4)	21.7 (11.3)	6.44 **	1.84 to 11.05

Mean differences are reported for between group interaction effect between exercise and education groups and for the effect of time for the groups combined (time effect). For primary outcome measures, improvement is indicated by a decrease in scores. For secondary outcome measures, improvement is indicated by an increase. * *p* < 0.05, ** *p* < 0.01. MNSI: Michigan Neuropathy Screening Instrument.

## Data Availability

The data presented in this study are available on request from the corresponding author.

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
