# Peer review of "The Effect of Structured Exercise Compared with Education on Neuropathic Signs and Symptoms in People at Risk of Neuropathic Diabetic Foot Ulcers: A Randomized Clinical Trial"

_medicina, 2021, doi:10.3390/medicina58010059_

Round 1

Reviewer 1 Report

Title of work: The effect of structured exercise compared with education on neuropathic 1 signs and symptoms in people at risk of neuropathic diabetic foot ulcers: a 2 randomized clinical trial

Authors: Byron M Perrin, Jessica Southon, Jane McCaig, Isabelle Skinner, Timothy C Skinner, Michael I C Kingsley

Journal (planned for publication): Medicina

Review of manuscript:

This study is well-designed. The researchers were chosen the clinical parameters properly. The number of patients was enough to obtain data from this study. The authors’ method is adequate and the statistical analyses seems to be correct. Results were given clearly with sufficient tables and data analysis.

Questions and remarks:

  • The diagnostic measurements must be appropriate to examine the patients. According to the ADA and EASD guidelines, diabetic neuropathy is a diagnosis of exclusion. Non-diabetic neuropathies can occur in diabetics and can be treated with special measures. The diagnosis of DSPN is primarily clinical in nature. Other ethiological factors of neuropathy were excluded by the diabetic patients?
  • What criteria did the authors use to determine the exact type of diabetes that ruled out any 1 types of diabetics with antibody testing?
  • Did the medication of patients, or any insulin therapy, play a role in planning and performing exercise?
  • Have patients received special medication (alpha-lipoic acid, benfotiamin, pregabalin, etc.) in the care of diabetic neuropathy?

It would be worthwhile to supplement the methods and the discussion of the study with answers to this questions and comments.

Author Response

The authors thank the Reviewer for their considered comments. Some revisions have been made on the basis of these comments and we believe the manuscript is improved.

1.1 The diagnostic measurements must be appropriate to examine the patients. According to the ADA and EASD guidelines, diabetic neuropathy is a diagnosis of exclusion. Non-diabetic neuropathies can occur in diabetics and can be treated with special measures. The diagnosis of DSPN is primarily clinical in nature. Other ethiological factors of neuropathy were excluded by the diabetic patients?

1.1 Response: The diagnosis of peripheral neuropathy was based on the Michigan Neuropathy Screening Instrument. We agree with the reviewer that it is possible that there are causes of peripheral neuropathy other than diabetes. These were not formally excluded in this study. This limitation has been explained more clearly in the limitations section

1.1 Action: Text in limitations sections added: “Due to the short intervention duration Hb1Ac measurements were not included in this study, and as the diagnosis of peripheral neuropathy was based on clinical testing, it is possible that diabetes was not the cause of neuropathy in all participants.”

1.2 What criteria did the authors use to determine the exact type of diabetes that ruled out any 1 types of diabetics with antibody testing?

1.2 Response: The diagnosis of diabetes was confirmed by the participants General Medical Practitioner. This included information about type of diabetes, which is reported in Table 1.

1.2 Action: No change to text

1.3 Did the medication of patients, or any insulin therapy, play a role in planning and performing exercise?

1.3 Response: Exercise prescription was individualised and considered medical history. This has been made clear in paragraph 2, page 6.

1.3 Action: Text changed in paragraph 2, page 6: “The program was individually prescribed with full consideration of medical history and was undertaken in a group environment of no more than eight participants.”

1.4 Have patients received special medication (alpha-lipoic acid, benfotiamin, pregabalin, etc.) in the care of diabetic neuropathy?

1.4 Response: No special medication was recorded for the treatment of peripheral neuropathy. We agree this should be made more clear.

1.4 Action: Text changed in paragraph 2, page 5: “Exclusion criteria were: diagnosis of painful neuropathy, use of medication for the treatment of peripheral neuropathy, active foot ulcer,…”

Reviewer 2 Report

The article is well written and offers ideas of originality, however the small sample size and the absence of some laboratory data, such as glycosylated hemoglobin, strongly limit the generalization of the results.  Some points deserve to be better clarified:

  • Please add the references at the end of periods.

Introduction

  • Line 42: The authors should briefly clarify how the diagnosis of peripheral neuropathy is currently made.
  • Line 42: Some authors have recently described an involvement of plantar nerves, please insert a brief comment with the reference. “Galiero R, Ricciardi D, Pafundi PC, Todisco V, Tedeschi G, Cirillo G, Sasso FC. Whole plantar nerve conduction study: A new tool for early diagnosis of peripheral diabetic neuropathy. Diabetes Res Clin Pract. 2021 Jun;176:108856. doi: 10.1016/j.diabres.2021.108856. Epub 2021 May 7. PMID: 33965449.”

Discussion

  • Line 189: The authors should briefly describe the drugs, and their side effects, currently used to relief the pain in a condition of peripheral neuropathy.
  • Line 190: The authors should briefly describe the advantages of a lifestyle intervention in diabetic patients.
  • Line 230: Authors should briefly provide hypotheses to motivate this result.

Author Response

The authors thank the Reviewer for their considered comments. Some revisions have been made on the basis of these comments and we believe the manuscript is improved.

2.1 Please add the references at the end of periods.

2.1 Response: The authors have checked the in-text citations and all are within periods in line with author instructions. Please let us know if we have misunderstood this comment.

2.1 Action: No changes.

2.2 Line 42: The authors should briefly clarify how the diagnosis of peripheral neuropathy is currently made.

2.2 Response: The diagnosis of peripheral neuropathy was based on the Michigan Neuropathy Screening Instrument. The diagnosis tool used is explained in paragraph 2, page 5.

2.2 Action: No text changes has been made in the methods, however possible limitations to using this clinically-based tool have been added (paragraph 2, p17): “Due to the short intervention duration Hb1Ac measurements were not included in this study, and as the diagnosis of peripheral neuropathy was based on clinical testing, it is  possible that diabetes was not the cause of neuropathy in all participants.”

2.3 Line 42: Some authors have recently described an involvement of plantar nerves, please insert a brief comment with the reference. “Galiero R, Ricciardi D, Pafundi PC, Todisco V, Tedeschi G, Cirillo G, Sasso FC. Whole plantar nerve conduction study: A new tool for early diagnosis of peripheral diabetic neuropathy. Diabetes Res Clin Pract. 2021 Jun;176:108856. doi: 10.1016/j.diabres.2021.108856. Epub 2021 May 7. PMID: 33965449.”

2.3 Response: The authors have reviewed this reference. We feel the most appropriate way to consider the information in in the context of more sensitive diagnostic tools to diagnose peripheral neuropathy.

2.3 Action: The following two sentences have been added in the limitations section: “The benefit of exercise and lifestyle interventions in people with less severe peripheral neuropathy should be explored in future research. Utilisation of more sensitive diagnostic tools, such as nerve conduction studies, might allow the researchers to explore the mechanisms that explain the responses to lifestyle interventions [39].”

2.4 Line 189: The authors should briefly describe the drugs, and their side effects, currently used to relief the pain in a condition of peripheral neuropathy.

2.4 Response: None of the participants had a formal diagnosis of painful neuropathy. As such a general description of drugs to treat painful neuropathy is not in the scope of the current paper.

2.4 Action: The text has been changed in the methods to make the neuropathy characteristics of the sample more clear (paragraph 2, page 5): “Exclusion criteria were: diagnosis of painful neuropathy, use of medication for the treatment of peripheral neuropathy, active foot ulcer,…”

2.5 Line 190: The authors should briefly describe the advantages of a lifestyle intervention in diabetic patients.

2.5 Response: The authors thank the review for seeking more explanation about this. We agree this is needed.

2.5 Action: The following sentence has been added to paragraph 2, page 16 in the discussion: “Lifestyle interventions may reduce reliance on medications through metabolic benefits associated with body mass control (e.g. blood pressure, lipoprotein balance) as well as resulting in musculoskeletal functional benefits [34].”

2.6 Line 230: Authors should briefly provide hypotheses to motivate this result.

2.6 Response: The hypothesis has been included in the aims section.

Round 2

Reviewer 2 Report

The authors have provided changes that have allowed to improve the readability of the text and to clarify some points.